# The PitA protein contributes to colistin susceptibility in *Pseudomonas aeruginosa*

**Mareike B. Erdmann**[ID]**, Paul P. Gardner**[ID]**, Iain L. Lamont**[ID]*

Department of Biochemistry, University of Otago, Dunedin, New Zealand

* iain.lamont@otago.ac.nz

## Abstract

*Pseudomonas aeruginosa* is an opportunistic pathogen that causes a wide range of problematic infections in individuals with predisposing conditions. Infections can be treated with colistin but some isolates are resistant to this antibiotic. To better understand the genetic basis of resistance, we experimentally evolved 19 independent resistant mutants from the susceptible laboratory strain PAO1. Whole genome sequencing identified mutations in multiple genes including *phoQ* and *pmrB* that have previously been associated with resistance, *pitA* that encodes a phosphate transporter, and *carB* and *eno* that encode enzymes of metabolism. Individual mutations were engineered into the genome of strain PAO1. Mutations in *pitA*, *pmrB* and *phoQ* increased the minimum inhibitory concentration (MIC) for colistin 8-fold, making the bacteria resistant. Engineered *pitA/phoQ* and *pitA/pmrB* double mutants had higher MICs than single mutants, demonstrating additive effects on colistin susceptibility. Single *carB* and *eno* mutations did not increase the MIC suggesting that their effect is dependent on the presence of other mutations. Many of the resistant mutants had increased susceptibility to β-lactams and lower growth rates than the parental strain demonstrating that colistin resistance can impose a fitness cost. Two hundred and fourteen *P. aeruginosa* isolates from a range of sources were tested and 18 (7.8%) were colistin resistant. Sequence variants in genes identified by experimental evolution were present in the 18 resistant isolates and may contribute to resistance. Overall our results identify *pitA* mutations as novel contributors to colistin resistance and demonstrate that resistance can reduce fitness of the bacteria.

## Introduction

The Gram negative bacterium *Pseudomonas aeruginosa* causes severe infections in patients with predisposing conditions such as bronchiectasis due to cystic fibrosis or other chronic forms of lung disease and in people who are hospitalised or immunocompromised, causing a burden for public health [1]. *P. aeruginosa* is a member of the ESKAPE group of pathogens that, because of high levels of antibiotic resistance, are problematic to treat [2, 3] and infections with antibiotic resistant *P. aeruginosa* have become a major issue for intensive care units (ICU) at hospitals, especially for immunocompromised patients [4, 5].

**Data Availability Statement:** All relevant data are within the paper and its Supporting Information files.

**Funding:** This work was supported by a grant from the Health Research Council of New Zealand (17/372) (www.hrc.govt.nz) to IL. The funders had no

role in study design, data collection and analysis, decision to publish, or preparation of the manuscript.

**Competing interests:** The authors have declared that no competing interests exist.

Management of *P. aeruginosa* infections involves antibiotic monotherapy or combination therapy and can involve antibiotics from a range of different classes including β-lactams, aminoglycosides, fluoroquinolones and polymyxins [1, 6]. Colistin belongs to the polymyxin class of antibiotics, of which polymyxin B and polymyxin E (colistin) are used for treatment [7]. Colistin is on the list of essential medicines from the WHO [8]. It has been considered as a 'last line of defence' antibiotic reserved for treatment of infections by multidrug resistant Gram negative pathogens due to its side effect of rapid deterioration of kidney function [9, 10]. However the development of an aerosilised form that can be inhaled directly to treat lung infections has reduced toxic side effects and it is commonly used in treatment of patients with cystic fibrosis or other forms of bronchiectasis [11]. An increased use of colistin has resulted in the emergence of colistin-resistant *P. aeruginosa*, with up to 10% of *P. aeruginosa* being resistant in some studies [12–15]. Resistance to an antibiotic can alter susceptibility to other antibiotics and can also affect bacterial fitness [16–18]. Knowledge of whether colistin resistance affects susceptibility to other antibiotics or affects bacterial fitness is limited but could help to improve treatment options for patients with infections of colistin-resistant *P. aeruginosa*.

Polymyxins work by interacting with lipopolysaccharide (LPS) in the Gram negative cell envelope, resulting in cell death due to disruption of the outer membrane and loss of its barrier function as well as interaction with nascent LPS at the cytoplasmic membrane and generation of hydroxyl radicals [19–21]. In *P. aeruginosa* resistance commonly involves modification of lipid A of lipopolysaccharide by attachment of 4-amino-4-deoxy-L-arabinose (L-Ara4N) or phosphoethanolamine (pEtN), reducing the negative charge of the LPS and consequently its affinity of colistin. Mutations in the two component systems PhoPQ, PmrAB, ParRS and CprRS can all result in increased expression of the *arnBCADTEF-ugd (pmrHFJKLME)* operon that encodes enzymes required for attachment of L-Ara4N to lipid A and plays a key role in colistin resistance [22] [reviewed in [19, 21, 23]]. Increased expression of the *eptA* gene leads to the addition of the cationic substance pEtN to lipid A resulting in reduced colistin affinity [24]. In other species resistance to colistin can also arise through acquisition of resistance genes of the *mcr* family, located on mobile genetic elements [25]. *Mcr* genes also confer resistance by causing addition of pEtN to lipid A. Although the above mechanisms for colistin resistance are relatively well understood, understanding of the genetic basis of colistin resistance and its effects on fitness or susceptibility to other antibiotics is incomplete. One approach to better understand the basis and consequences of colistin resistance is to carry out experimental evolution studies. Mutants of *P. aeruginosa* that were resistant to colistin in liquid culture have been developed in two studies [26, 27] and both identified mutations in *pmrB*, *lpxC* and *mutS*, while mutations in other genes were observed in only one of the two studies. The aims of this study were to identify colistin resistance-associated genes using an agar-based experimental evolution method and then quantify the impacts of key mutations individually and in combination in reducing susceptibility to colistin; and to determine the effects of colistin resistance on fitness and on susceptibility to other antibiotics.

## Results

### Experimental evolution of colistin-resistant *P. aeruginosa*

The colistin-sensitive *P. aeruginosa* strain PAO1 was exposed to increasing concentrations of colistin using agar plates containing concentration gradients of the antibiotic, with up to six cycles of selection. This stepwise procedure resulted in 19 independent experimentally evolved mutants. The MIC was measured for each (Fig 1A). The $MIC_{50}$ of the parental strain was 0.5 mg/L, which was below the breakpoint for colistin resistance (R $\geq$ 4 mg/L) [28]. All of the evolved mutants had reduced susceptibility to colistin compared to strain PAO1 in both

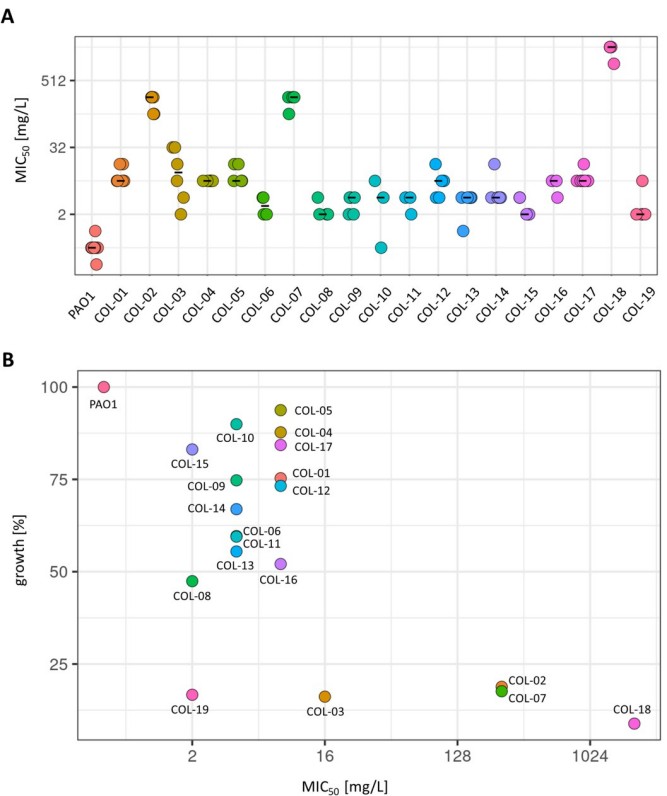

**Fig 1. Colistin MICs and growth of experimentally evolved mutants. A.** $MIC_{50}$ of the experimentally evolved mutants. Biological replicates are displayed as individual data points, with black lines indicating the median values. **B.** Growth compared to $MIC_{50}$ for each mutant. Growth (area under the curve) was the average of three biological replicates and is shown as a percentage of growth of parental strain PAO1.

$MIC_{50}$ (Fig 1A) and $MIC_{90}$ (S1 Fig) measurements. The $MIC_{50}$ for the mutants ranged between 4 and 2048 mg/L. The $MIC_{50}$ values were not directly correlated with the number of selection cycles. Different mutants obtained after three, four or six selection cycles all had an $MIC_{50}$ of 8 mg/L (S1 Table in S1 File). Conversely, after five selection cycles the $MIC_{50}$ value in one evolved mutant was 2 mg/L while in another it was 8 mg/L.

## Colistin resistance can result in reduced growth

Previous studies have reported an impact of colistin resistance on growth in *Salmonella enterica* and *Acinetobacter baumanii* bacteria [29, 30]. The growth of each evolved mutant in broth culture was measured to identify potential correlations between colistin resistance and growth (Fig 1B and S1 Table in S1 File). All 19 mutants had impaired growth relative to the parental strain PAO1, with 6 mutants having a growth reduction of more than 50% (S1 Table in S1 File). Notably, the growth of all evolved mutants with $MIC_{50}$ values of $\geq 256$ mg/L was significantly reduced (p < .001) compared to strain PAO1. Interestingly, some mutants with significant growth reductions had relatively low $MIC_{50}$s ($\leq 4$ mg/L).

## Multiple genes were mutated in different evolved mutants

Comparison of the genome sequences of the 19 evolved mutants with that of the parental strain PAO1 identified between one and four mutated genes in each mutant (S2 Table in S1 File). A total of seven genes, and two further genes in the same operons, were mutated in more

than one mutant (Table 1). All mutations in these genes were insertions, deletions or amino acid substitutions that were predicted by PROVEAN, a protein-variant analysis tool, to significantly affect protein function (S3 Table in S1 File).

Mutations had arisen most frequently in *pmrB* [15 mutants], a two component regulatory system that is known to contribute to colistin resistance of *P. aeruginosa* [26, 31, 32]. The second most frequent mutations were in the *pitA* gene, that has recently been shown to encode a phosphate transporter [33]. Mutations in the *pitA* gene, and also in the *carB* and *eno* genes, have not previously been reported in association with colistin resistance. All mutations in the *carB* and *eno* genes were deletion mutations resulting in a frameshift and therefore a likely loss of function in the respective proteins. Four of the evolved mutants had mutations in the *speDE2* operon that encodes enzymes for polyamine synthesis [34].

## Prevalence of sequence variants in resistance-associated genes in clinical, environmental and animal isolates

The frequency of colistin resistance and the prevalence of mutations in the most frequently mutated genes was assessed in isolates of clinical, environmental and animal origin. A total of 214 *P. aeruginosa* isolates, representing the phylogenetic breadth of the species (S2 Fig), was analysed. The $MIC_{50}$ of colistin was determined for the isolates, identifying 18 isolates (8.4%) as being colistin resistant (S4 Table in S1 File). Thirteen of the resistant isolates were of clinical origin, and 5 were from the general environment. BLAST analysis showed that none of the 214 isolates contained *mcr* genes, that can be acquired by horizontal gene transfer and are known contributors to colistin resistance [25]. Resistant isolates were widely distributed on the phylogenetic tree, indicating that colistin resistance had arisen independently on multiple occasions and was not monophyletic.

Sequence variants in the genes of interest were analysed for the 214 isolates using PROVEAN. Variations with significant scores were identified in 8 of the 9 genes examined (Table 2). The *phoQ* gene had significant sequence variants in the highest number of isolates, and *speD2* had no significant variants. Eight of the 18 colistin resistant isolates had a significant variant in at least one of *pmrAB*, *pitA*, *phoQ*, *speE2* and PA5005. With the exception of *pitA*, mutations in those genes have been associated with colistin resistance in one or more previous studies [26, 32]. No significant variants in the colistin resistant isolates were observed in *carB* or *eno*. Some of the significant variants (*phoQ*<sub>Y85F</sub>, *phoQ*<sub>S300R</sub>, *speE2*<sub>P326A</sub> and PA5005<sub>D23E</sub>) were present in both colistin-resistant and colistin-sensitive isolates. This indicated that these

**Table 1. Genes mutated in multiple evolved colistin-resistant mutants*.**

| Protein function | gene | PAO1 locus | Number of mutants [maximum 19] |
|---|---|---|---|
| two component regulator *pmrAB* | *pmrB* | PA4777 | 15 |
| | *pmrA* | PA4776 | 1 |
| phosphate transporter | *pitA* | PA4292 | 7 |
| two component regulator *phoPQ* | *phoQ* | PA1180 | 3 |
| carbamoyl phosphate synthase | *carB* | PA4756 | 3 |
| polyamine synthesis | *speE2* | PA4774 | 3 |
| | *speD2* | PA4773 | 1 |
| carbamoyl transferase | PA5005 | PA5005 | 2 |
| enolase | *eno* | PA3635 | 2 |

*Genes mutated in 2 or more of the independently evolved mutants are listed, along with mutated operonic genes

**Table 2. Prevalence of significant sequence variants in colistin-resistant and -sensitive *P. aeruginosa* isolates.**

|        | Resistant      | Sensitive      |
|--------|----------------|----------------|
| PmrB   | 2 [11.1 %]*    | 14 [7.1 %]     |
| PmrA   | 1 [5.6 %]      | 7 [3.6 %]      |
| PitA   | 4 [22.2 %]     | 5 [2.6 %]      |
| PhoQ   | 4 [22.2 %]     | 22 [11.2 %]    |
| CarB   | 0              | 5 [2.6 %]      |
| SpeE2  | 1 [5.6 %]      | 11 [5.6 %]     |
| SpeD2  | 0              | 0              |
| PA5005 | 2 [11.1 %]     | 12 [6.1 %]     |
| Eno    | 0              | 2 [1.0 %]      |

*Sequence variants with a significant PROVEAN score

single gene mutations were not sufficient to confer colistin resistance but may do so in combination with other mutations.

## Effects of individual mutations on colistin susceptibility

Most of the experimentally evolved colistin resistant mutants had multiple mutations. Two different approaches were used to quantify the effects on mutations in individual genes. Firstly, the effects of transposon insertions were assessed using mutants from a well-characterized transposon library [35]. Only an insertion near the middle of *pitA* resulted in an increase in $MIC_{50}$, from 0.5 mg/L to 4 mg/L (Fig 2A). A second insertion near the start of the *pitA* gene did not impact susceptibility, likely because expression of the remaining portion of *pitA* was sufficient to retain protein function [36]. Transposon mutations in *phoQ*, *pmrB*, *eno*, *carB*, *speE2* and PA5005 did not increase colistin $MIC_{50}$, and mutants were not available for *pmrA* and *speD2*.

The impact of mutations that arose in the experimental evolution study was then quantified. Four single gene mutations were chosen based on the frequency of occurrence in experimentally evolved mutants, reports in the scientific literature and the results of the loss of function in the transposon mutants: $pitA_{\Delta 11 \text{ bp del}}$, $phoQ_{V260G}$, $carB_{\Delta 5 \text{ bp del}}$ and $pmrB_{M292T}$. Each of these mutations was engineered into the genome of strain PAO1. The $pmrB_{M292T}$ mutation that was the most frequent mutation in the evolved mutants resulted in a 4-fold increase in $MIC_{50}$ to 2 mg/L (Fig 2B). The $pitA_{\Delta 11 \text{ bp del}}$ mutation increased the $MIC_{50}$ to 4 mg/L. The highest increase in IC50 occurred with the PAO1 $phoQ_{V260G}$ mutant that had an $MIC_{50}$ of 8 mg/L. The PAO1 $carB_{\Delta 5 \text{ bp del}}$ mutant had the same $MIC_{50}$ as PAO1, consistent with the *carB* transposon mutant having no effect on colistin susceptibility.

## Different point mutations in *pmrB* impact colistin tolerance differently

A variety of different mutation sites within the *pmrB* gene arose in the experimentally evolved mutants, with four evolved mutants only having a mutation (Δ12 bp del, V136E, R155H or P169A) in *pmrB*. These mutants and the mutant engineered to have only the *pmrB* M292T mutation had 4- to 16-fold increases in colistin $MIC_{50}$ compared to PAO1 (Fig 2B). The M292T that arose most frequently in the evolved mutants (4/19) did not cause the largest increase in $MIC_{50}$. The *pmrB* mutations were not in spatial proximity on predicted PmrB protein structure (S3 Fig), suggesting that they do not target a single active site in the PmrB protein.

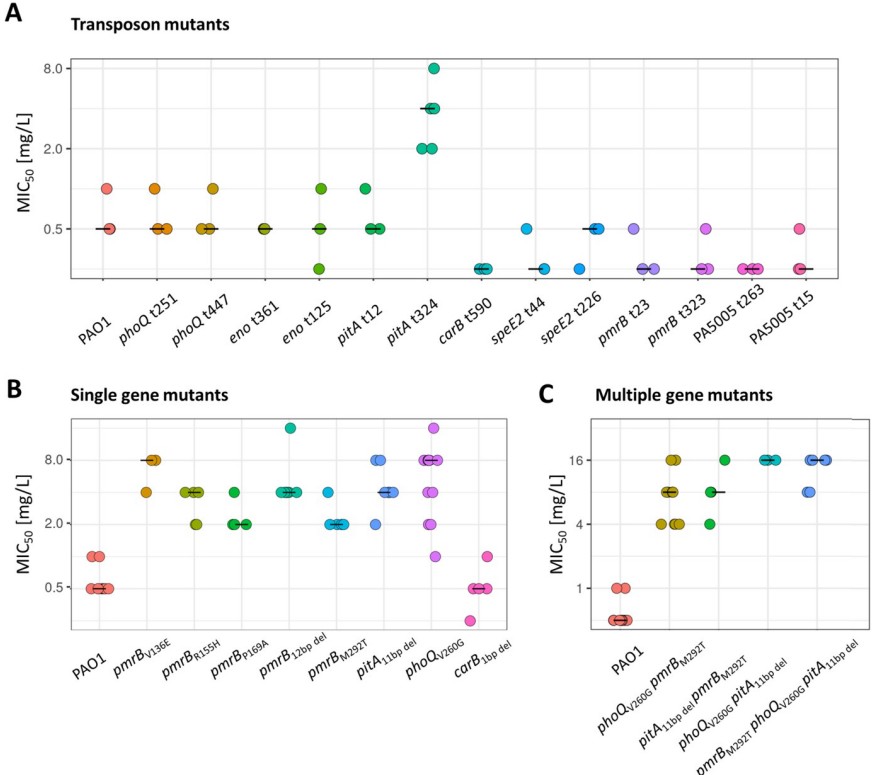

**Fig 2. Colistin MICs of transposon and engineered mutants. A.** $MIC_{50}$ of mutants with transposon insertions. The transposon insertion sites are shown. **B.** $MIC_{50}$ of mutants with mutations engineered in single genes and of evolved mutants with single mutations in *pmrB*. **C.** $MIC_{50}$ of multi-gene mutants. Biological replicates are displayed as individual data points, with black lines indicating the median values.

## Combinations of *pitA* with *phoQ* or *pmrB* mutations further reduce colistin susceptibility

All of the evolved mutants with a *pitA* mutation also had a mutation in *pmrB* or *phoQ* (S2 Table in S1 File). To study potential epistatic effects between mutations, double and triple mutants were engineered using the $pitA_{\Delta 11\ bp\ del}$, $pmrB_{M292T}$ and $phoQ_{V260G}$ alleles. The *pitA pmrB* and *pitA phoQ* double mutants had higher $MIC_{50}$ values than the corresponding single gene mutants, whereas the *pmrB phoQ* double mutant had the same $MIC_{50}$ as the *pmrB* and *phoQ* single mutants (Fig 2C). The *pitA phoQ pmrB* triple mutant had the same $MIC_{50}$ as the *pitA phoQ* double mutant. These findings show that *pitA* mutations alter susceptibility to colistin through a different mechanism to *phoQ* and *pmrB* mutations, which cause L-Ara4N to be attached to lipid A.

## Mutations that reduce susceptibility to colistin can also result in reduced growth

The impact of single gene mutations on bacterial growth was measured. An engineered *carB* mutation and all of the different point mutations in *pmrB* resulted in reduced growth compared to PAO1 (S5 Table in S1 File). A correlation between colistin tolerance and growth reduction was observed for the *pmrB* mutations, as the mutations $pmrB_{P169A}$ and $pmrB_{M292T}$ that gave the lowest $MIC_{50}$ values had the least effects on growth of all *pmrB* mutations. No reduction in growth was observed in engineered *pitA* and *phoQ* mutants or in *eno*, *pitA*, *speE2*

or PA5005 transposon mutants (S5 Table in S1 File). However, transposon mutations in *phoQ* and *carB* resulted in significantly reduced growth compared to strain PAO1. Significant growth reductions compared to PAO1 were also observed in the engineered double mutants although not the triple mutant (S5 Table in S1 File).

## Mutations associated with colistin resistance can impact susceptibility to other classes of antibiotics

The MICs of antibiotics in classes commonly used to treat *P. aeruginosa* infections were measured for the experimentally evolved colistin resistant mutants. Ten of the evolved mutants had slightly increased susceptibility to ceftazidime, and 17 to meropenem, indicating that colistin resistance can increase susceptibility to β-lactam antibiotics (Table 3) and consistent with an earlier study [18]. There was no clear correlation between MIC and the mutated genes in each mutant. Most of the evolved mutants had the same MIC for tobramycin as the parental PAO1 strain, though four mutants had increased susceptibility. The MIC for ciprofloxacin was generally unaffected by the acquisition of mutations associated with colistin resistance.

## Discussion

Colistin is an important antibiotic in the treatment of *P. aeruginosa* infections but the genetic basis of colistin resistance is only partially understood. Our results demonstrate the involvement of the PitA phosphate transporter and quantify the effects of *pitA* mutations in combination with other resistance mechanisms in determining levels of colistin susceptibility. They also show that colistin resistance can come at a cost, reducing fitness and increasing susceptibility to β-lactam antibiotics.

Resistance to colistin arises through modifications of LPS that reduce the affinity of the negatively charged LPS for the positively charged colistin molecules. Previously characterised modifications involve addition of L-Ara4N to lipid A, due to mutations in *phoQ* or *pmrB* that increase expression of the *pmrHFJKLME/arnBCADTEF-ugd arn* operon (Fig 3B) [31, 37], and addition of pEtN catalysed by *mcr* genes [24, 38]. These modifications likely reduce the affinity of LPS for colistin. Our findings suggest two additional mechanisms for modification of LPS. Both the engineered and transposon *pitA* mutations, in the absence of any other mutations, resulted in an increase of $MIC_{50}$ for colistin by 8-fold (Fig 2A and 2B). That change of $MIC_{50}$ was sufficient for strain PAO1 to become resistant. The PitA protein has recently been identified as playing a role in phosphate uptake [33]. Phosphate groups present in the inner core of LPS can contribute to colistin binding [31, 39]. Reduced availability of phosphate may reduce the amount of negatively charged phosphate in the LPS (Fig 3C), reducing affinity for the positively charged colistin molecules and hence reducing the $MIC_{50}$. However colistin's mode of action is complex and involves disruption of both the inner and outer membranes as well as generation of hydroxyl radicals [20, 21]. Mutations in *pitA* alter the proton motive force [33] and so resistance associated with *pitA* mutations may also be due to the changed membrane potential reducing cellular entry of the positively charged colistin molecules, or reducing the formation of hydroxyl radicals. Intriguingly, deletion of *pitA* did not affect colistin resistance in strain PA14 [33], perhaps because the amount of phosphate in LPS varies between isolates [39]. The effects of *pitA* mutations in other isolates remain to be determined. It will also be of interest to determine whether the amount of phosphate in the growth medium influences susceptibility to colistin.

A second potential mechanism for modification of LPS was revealed by the finding that five of the experimentally evolved mutants had mutations in PA5005 or *carB*. The CarB enzyme synthesises carbamoyl phosphate [40] and the enzyme encoded by the PA5005 gene

**Table 3. Susceptibility of evolved mutants to other antibiotics.**

| Mutant | Meropenem | Ceftazidime | Tobramycin | Ciprofloxacin |
|--------|-----------|-------------|------------|---------------|
| | β-lactam | β-lactam | Aminoglycoside | Fluoroquinolone |
| PAO1 | 1* | 1 | 0.5 | 0.03125 |
| COL-01 | 0.5 | 1 | 0.5 | 0.03125 |
| COL-02 | 0.5 | 0.5 | 1 | 0.03125 |
| COL-03 | 0.25 | 0.5 | 0.125 | 0.03125 |
| COL-04 | 1 | 1 | 0.5 | 0.03125 |
| COL-05 | 1 | 1 | 0.5 | 0.03125 |
| COL-06 | 0.25 | 0.5 | 0.5 | 0.03125 |
| COL-07 | 0.5 | 0.5 | 0.5 | 0.03125 |
| COL-08 | 1 | 1 | 0.5 | 0.03125 |
| COL-09 | 0.5 | 0.5 | 1 | 0.03125 |
| COL-10 | 0.5 | 0.5 | 0.5 | 0.03125 |
| COL-11 | 0.5 | 0.5 | 0.5 | 0.03125 |
| COL-12 | 0.125 | 0.25 | 0.125 | 0.03125 |
| COL-13 | 0.5 | 1 | 0.5 | 0.03125 |
| COL-14 | 1 | 1 | 0.5 | 0.03125 |
| COL-15 | 0.5 | 1 | 0.5 | 0.03125 |
| COL-16 | 0.5 | 0.5 | 0.125 | 0.03125 |
| COL-17 | 0.25 | 0.25 | 0.25 | 0.03125 |
| COL-18 | 0.5 | 1 | 0.25 | 0.0625 |
| COL-19 | 0.5 | 1 | 0.5 | 0.03125 |

*MIC values are shown. All values are the median of at least 3 biological replicates. Clinical breakpoints for resistance are: meropenem $\geq$ 8 mg/L; ceftazidime $\geq$ 8 mg/L; tobramycin $\geq$ 2 mg/mL; ciprofloxacin $\geq$ 0.5 mg/L [28].

Red, decrease in susceptibility relative to strain PAO1; Ochre, increase in susceptibility relative to strain PAO1.

(provisionally named WapO) is likely responsible for the addition of carbamoyl groups to the inner core of LPS [39]. It therefore seems likely that absence of carbamoyl groups from the LPS, due to mutations in PA5005 or *carB*, reduces the affinity of the LPS for colistin. Mutations in *carB* or PA5005 did not by themselves increase the $MIC_{50}$ (and in some cases, reduced the $MIC_{50}$) indicating that they must act epistatically in combination with other mutations to reduce susceptibility to colistin.

The identification of *phoQ* and *pmrB* mutations in evolved mutants and the impacts of engineered single gene mutations on colistin susceptibility were consistent with observations of previous studies [26, 27, 31, 41]. The $MIC_{50}$ of the *phoQ pmrB* double mutant was no higher than that of the *phoQ* single gene mutant, consistent with both mutations being part of the same resistance mechanism. However, *pitA phoQ* and *pitA pmrB* double mutants had higher $MIC_{50}$s than the respective single gene mutants consistent with the *pitA* mutation acting through a different resistance mechanism than the *phoQ* and *pmrB* mutations. It would be of interest to determine if *pitA* mutations also increase the $MIC_{50}$ of bacteria with mutations in other resistance-associated genes, such as *carB* or *eno*.

Point mutations in *pitA*, as well as in *phoQ* or *pmrB*, were sufficient to make strain PAO1 resistant to colistin (Fig 2). However sequence variants predicted to affect the function of these proteins, and variants in other genes associated with resistance, were present in *P. aeruginosa* isolates that were susceptible to colistin (Table 2) showing that they are not necessarily sufficient to cause resistance. Furthermore, over half of the 18 resistant isolates examined did not have predicted function-altering variants in any of the genes that had undergone mutation in

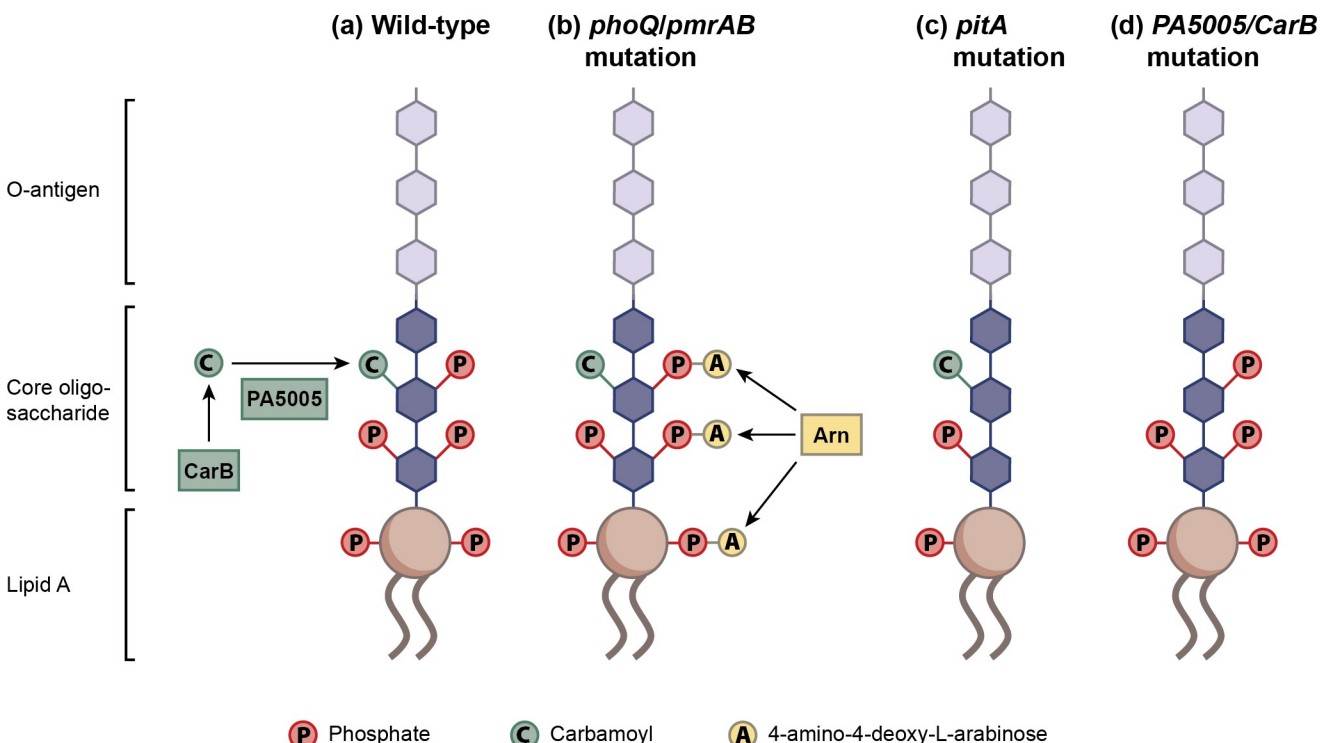

**Fig 3. Schematic model for changes to LPS that reduce susceptibility to colistin. A.** In wild-type bacteria, negatively charged carbamoyl and phosphate groups interact with colistin, concentrating the antibiotic at the cell surface. CarB catalyses synthesis of carbamoyl phosphate, the substrate for the PA5005 (WapO) carbamoyl transferase enzyme that adds a carbamoyl group to a heptose residue in the core oligosaccharide. **B.** Mutations in *phoQ* or *pmrAB* result in increased synthesis of Arn proteins that catalyse addition of L-Ara4N to phosphate groups of lipid A, masking the negative charge and reducing affinity for colistin. **C.** Mutations in *pitA* reduce the availability of phosphate in lipid A and for decoration of heptose sugars in the inner core of lipopolysaccharide, reducing affinity for colistin. **D.** Mutations in *carB* or PA5005 prevent incorporation of carbamoyl groups to the inner core, reducing affinity for colistin.

multiple experimentally-evolved colistin-resistant mutants. These findings further emphasise the complex and polygenic nature of colistin resistance, with the effects of individual mutations being influenced by the genetic background. More detailed investigation of clinical and environmental isolates and improved understanding of gene-gene interactions will improve understanding of colistin resistance in *P. aeruginosa* and may help identify alternative pathways that increase colistin $MIC_{50}$. It may also enable prediction of the likely effectiveness of colistin treatment following genome sequencing of infecting bacteria [42, 43].

Four of the experimentally evolved mutants had mutations in *spe* genes that encode enzymes for polyamine synthesis. Under conditions of cation-limitation, cell-surface polyamines can substitute for $Mg^{2+}$ ions to stabilise LPS and may reduce susceptibility to polymixins. In this study, mutants were evolved and MICs determined in cation-adjusted MH medium that contains approximately 0.5 mM of $Mg^{2+}$ ions. These would normally repress expression of the *spe* genes [34]. The occurrence of mutations in *spe* genes in the evolved mutants may reflect the metabolic burden associated with their increased expression arising from mutations in *pmrAB* [44].

All of the experimentally evolved mutants grew less well than the parental PAO1 strain. The five experimentally evolved mutants with mutations in either *carB* or *eno* had the most striking differences, with growth reductions of more than 50% compared to PAO1. A clear relationship between the $MIC_{50}$ and growth was also observed for mutants carrying single mutations in *pmrB*, with the $pmrB_{V136E}$ mutation that resulted in the highest $MIC_{50}$ causing the most

significant growth reduction compared to PAO1. These findings demonstrate a cost to colistin resistance as noted previously for other species [29, 30]. Reduced fitness has potential to restore antibiotic susceptibility to a mixed population in the absence of antibiotic treatment [16] potentially slowing or reversing the emergence of resistance during chronic infections. Our findings indicate that as well as imposing a fitness cost, colistin resistance can increase susceptibility of some mutants to meropenem and ceftazidime and, in a smaller number of cases, to tobramycin. Of the 232 isolates that were tested in this study, 7.8% were resistant to colistin, a frequency that is similar to or higher than other studies [12–15]. The extent to which the frequency of resistance is affected by the balance between colistin usage and fitness costs in clinical, animal and environmental isolates remains to be determined.

Overall, our findings identify *pitA* as a novel contributor to colistin susceptibility in *P. aeruginosa* as well as quantifying the effects of interactions between *pitA* and *phoQ/pmrB* mutations. They also demonstrate that fitness costs are associated with colistin resistance, which may influence the emergence of resistance and the mutations associated with it during treatment. Fully understanding the genetic basis of colistin resistance has the potential to inform treatment based on personalized medicine approaches, including the use of alternative treatment options if mutations associated with colistin resistance are present.

## Material and methods

### Growth of bacteria

Bacteria used in this study are listed in S4 and S6 Tables in S1 File. They were maintained on LB agar and grown in LB broth at 37˚C with aeration unless otherwise stated. To quantify growth, overnight cultures of bacteria were diluted to $OD_{600\ nm}$ of 0.01 and portions (200 μL) aliquoted into 96-well tissue culture plates (Corning Inc, Corning, Corning, NY, USA). The plates were incubated in a BMG FLUOstar Omega microplate reader (37˚C, 200 rpm) for 24 hours and the $OD_{600\ nm}$ measured every 30 min. Growth was quantified as the area under the curve (AUC), calculated using the RStudio package '*growthcurver*' [45], as described previously [46]. Measurements of three biological replicates with four technical replicates each were used. The statistical significance of differences in AUC were calculated using one sample t-test, with the AUC of strain PAO1 set as hypothesized mean.

### Experimental evolution of mutants

Experimental evolution was performed using agar plates containing concentration gradients of colistin using a similar protocol to that described previously [46]. Gradient plates consisted of 15 mL cation-adjusted Mueller Hinton Broth (caMHB) agar dried at an angle as base layer, and 15 mL caMHB containing colistin as top layer. The starting concentration was 1 mg/L colistin in the top layer. Bacteria grown in antibiotic-free broth were inoculated onto the agar and incubated at 37˚C. An isolated single colony from each gradient plate was inoculated into antibiotic-free broth, incubated overnight at 37˚C, and the selection process repeated using a gradient plate with twice the concentration of colistin. The procedure was repeated until no single colonies were obtained. Passaging the sensitive strain PAO1 on non-antibiotic media did not result in mutations [46], indicating that the mutations obtained during experimental evolution were associated with antibiotic resistance.

### Antibiotic susceptibility testing

The minimal inhibitory concentration (MIC) of colistin was determined using broth microdilution in 96 well tissue culture plates with caMHB according to EUCAST v.11.0 guidelines [28].

*Escherichia coli* NCTC 13846 (MIC$_{50}$ 4 mg/L) was used as resistant control and *P. aeruginosa* ATCC 27853 (MIC$_{50}$ 0.5 mg/L) and *E. coli* ATCC 25922 (MIC$_{50}$ 0.125 mg/L) as sensitive controls. The MIC was tested in doubling concentrations ranging from 0.125 mg/L to 2048 mg/L. The optical density OD$_{600 \text{ nm}}$ was measured in the Omega microplate reader with the 50% inhibitory concentration being taken as the MIC$_{50}$ and the 90% inhibitory concentration as MIC$_{90}$. At least three biological replicates (two technical replicates each) were carried out for each mutant. MIC values for other antibiotics were determined using the doubling dilution method on MHB agar plates as described previously [46, 47].

### Mutant engineering

Mutants were engineered using the two step allelic exchange protocol [48, 49]. DNA fragments (∼2kb) carrying the mutations of interest were amplified from the genomes of evolved mutants using suitable primers (S7 Table in S1 File) and cloned into plasmid pEX18Gm [48]. The resulting plasmids were sequenced to confirm that only the intended mutation was present and then transformed into *E. coli* ST18 [50] and the mutations transferred into the genomes of recipient bacteria as described previously [49, 51]. Screening and confirmation of resulting *P. aeruginosa* mutants was carried out by sequencing PCR products generated using allele-specific screening primers (S7 Table in S1 File).

### Genome analysis

Accession numbers of *P. aeruginosa* analysed in this study are listed in S4 Table in S1 File. Genome sequences of isolates generated during this study were obtained following Illumina MiSeq sequencing as described previously [52–54]. The quality of the raw reads was controlled using *FastQC* v0.11.5 and *kraken* v2 [55] with low quality reads excluded using *trimmomatic* v.0.36 [56] (MAXINFO:50:0.8, LEADING:3, TRAILING:3 and MINLEN:36). Draft genome assemblies were generated using *SPAdes* v3.12.0 [57] with the default settings and the '-careful' option. Remaining genome sequences have been described previously [46, 51, 53, 58, 59].

The variant detector *Breseq* v0.33.1 [60] was used to identify mutations in the experimentally evolved mutants, using PAO1-Otago [46] as reference genome (accession number, SAMN11606715). Sequence assemblies were annotated using *prokka* v1.14.6 [61]. The protein variation effect analyzer PROVEAN v1.1.5 was used to predict the effects of sequence variants on protein function, with values below -2.5 considered to be significant [62]. The query sequences for the genes of interest were from the PAO1 strain (Accession: GCF_000006765.1) from pseudomonas.org [63]. The BLAST algorithm [64] was used to identify resistance genes in *P. aeruginosa* isolates.

### Supporting information

**S1 Fig. Colistin MIC$_{90}$ of experimentally evolved mutants.** Biological replicates are displayed as individual data points, with black lines indicating the median values.
(TIF)

**S2 Fig. Phylogenetic tree of colistin resistant and sensitive isolates.** 214 *P. aeruginosa* isolates of clinical, environmental and animal origin (S4 Table in S1 File) were phylogenetically analyzed using *parsnp* and the tree was visualized using iTOL. Inner circle: colistin-resistant isolates are shown in red, with colistin-sensitive isolates in black. The widespread distribution of colistin-resistant isolates demonstrates that resistance has arisen on multiple occasions. Outer circle: source of isolate. The reference strains PAO1 and PA14 and the location of

reference strain PA7 are included to visualize the variety of the isolates.
(TIF)

**S3 Fig. Changes to the PmrB protein.** The predicted PmrB protein structure (UniProtKB Q9HV31) was visualized using PyMOL. The structure of the protein is shown in green, with changes in evolved mutants highlighted as spheres. The M292T mutation in red was engineered into PAO1, mutations in cyan were present in evolved mutants with only one mutation and mutations in blue were present in evolved mutants with multiple mutations. For the Δ12 bp deletion mutation the first affected amino acid in the protein was coloured.
(TIF)

**S1 File. Supporting information and underlying data.** S1 Table, Growth and MICs for colistin of experimentally evolved mutants; S2 Table, Mutation profiles of the 19 experimentally evolved mutants; S3 Table, Mutations present in genes mutated in multiple evolved mutants, listed by gene; S4 Table, Colistin MICs of P. aeruginosa isolates of clinical, animal and environmental origin; S5 Table, Impacts of transposon and engineered mutations on growth; S6 Table, Laboratory strains of bacteria used in this study; S7 Table, DNA primers used in this study; S8 Table, Underlying data for Fig 1A; S9 Table, Underlying data for Fig 1B; S10 Table, Underlying data for Fig 2A; S11 Table, Underlying data for Fig 2B and 2C; S12 Table, Underlying data for Table 3.
(XLSX)

## Acknowledgments

We very gratefully acknowledge Kay Ramsay, Tim Kidd, Scott Bell, Claire Wainwright and Keith Grimwood for providing *P. aeruginosa* isolates and Scott Beatson and Roger Levesque and co-workers for providing genome sequences.

## Author Contributions

**Conceptualization:** Iain L. Lamont.

**Data curation:** Mareike B. Erdmann.

**Formal analysis:** Mareike B. Erdmann, Paul P. Gardner.

**Funding acquisition:** Iain L. Lamont.

**Investigation:** Mareike B. Erdmann.

**Methodology:** Mareike B. Erdmann.

**Project administration:** Iain L. Lamont.

**Supervision:** Paul P. Gardner, Iain L. Lamont.

**Validation:** Mareike B. Erdmann.

**Visualization:** Mareike B. Erdmann.

**Writing – original draft:** Mareike B. Erdmann, Iain L. Lamont.

**Writing – review & editing:** Mareike B. Erdmann, Paul P. Gardner, Iain L. Lamont.

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
