## [Decision Letter · Decision Letter 0]

5 Sep 2023

PONE-D-23-24646The PitA protein contributes to colistin susceptibility in Pseudomonas aeruginosaPLOS ONE

Dear Dr. Lamont,

Thank you for submitting your manuscript to PLOS ONE. After careful consideration, we feel that it has merit but does not fully meet PLOS ONE’s publication criteria as it currently stands. Therefore, we invite you to submit a revised version of the manuscript that addresses the points raised during the review process.

 Please address querries raised by the reviewer

We look forward to receiving your revised manuscript.

Kind regards,

Iddya Karunasagar

Academic Editor

PLOS ONE

Journal Requirements:

"We very gratefully acknowledge Kay Ramsay, Tim Kidd, Scott Bell, Claire Wainwright and Keith Grimwood for providing P. aeruginosa isolates and Scott Beatson and Roger Levesque and co-workers for providing genome sequences.

This work was supported by the Health Research Council of New Zealand (17/372)."

 "This work was supported by a grant from the Health Research Council of New Zealand (17/372) (www.hrc.govt.nz) to IL. The funders had no role in study design, data collection and analysis, decision to publish, or preparation of the manuscript."

Additional Editor Comments:

Please see the reviewer comments. Please revise addressing these.

Reviewers' comments:

Reviewer's Responses to Questions

**Comments to the Author**

1. Is the manuscript technically sound, and do the data support the conclusions?

Reviewer #1: Yes

2. Has the statistical analysis been performed appropriately and rigorously? 

Reviewer #1: Yes

3. Have the authors made all data underlying the findings in their manuscript fully available?

Reviewer #1: Yes

4. Is the manuscript presented in an intelligible fashion and written in standard English?

Reviewer #1: Yes

5. Review Comments to the Author

Reviewer #1: In the manuscript "The PitA protein contributes to colistin susceptibility in Pseudomonas aeruginosa" the authors demonstrate that the phosphate transporter PitA mutation contributes to the colistin resistance, and colistin resistance can reduce bacteria fitness. This highlights the important role of phosphate in resistance to antimicrobial compounds.

Overall, the experimental design is appropriated, and the results are clearly reported. About the Table 3, it may be helpful addition to analyze the impact of different single gene mutations on other antibiotic resistance.

It seems phosphate deficiency may play a role in LPS modification. I would like to know if authors considered exploring the contribution of PitA to polymyxins susceptibility in a phosphate deficient environment ?

6. PLOS authors have the option to publish the peer review history of their article (what does this mean?). If published, this will include your full peer review and any attached files.

Reviewer #1: No

---

## [Author Response · Author response to Decision Letter 0]

19 Sep 2023

Response to reviewer comments: manuscript PONE-D-23-24646

Reviewer and staff comments are shown in normal font, with our responses indicated by arrows. All references to line numbers in our responses refer to the numbering in the marked up revised manuscript. Our responses are also described in the "Response to reviewers" document.

Response to reviewer comments: Reviewer #1.

About the Table 3, it may be helpful addition to analyze the impact of different single gene mutations on other antibiotic resistance.

-> We carried out this analysis. There was no relationship evident between the single gene mutations and resistance to antibiotics other than colistin. As this result was not informative, we have not described it in the manuscript.

It seems phosphate deficiency may play a role in LPS modification. I would like to know if authors considered exploring the contribution of PitA to polymyxins susceptibility in a phosphate deficient environment ?

->This is an interesting suggestion, thank you. We have not explored the effects of phosphate deficiency on colistin susceptibility, which is beyond the aims and scope of this study. However we have now noted this in the discussion (lines 281-282) as a future research direction. 

Changes to meet journal requirements:

 "This work was supported by a grant from the Health Research Council of New Zealand (17/372) (www.hrc.govt.nz) to IL. The funders had no role in study design, data collection and analysis, decision to publish, or preparation of the manuscript."

-> We apologise for this error. We have now deleted the relevant sentence from the Acknowledgements section. We have included the new Funding Statement, using the wording recommended by the editorial staff, in our cover letter. Thank you in advance for changing the online submission form.

In your Data Availability statement, you have not specified where the minimal data set underlying the results described in your manuscript can be found. 

->We have incorporated underlying data into the supplementary tables.

->Data availability:

Figures 1A: underlying data are in Table S8 

Figure 1B: underlying data are in Table S1 and Table S9

Figure 2: underlying data are in Tables S10 and S11

Table 1: underlying data are in Table S2

Table 2: underlying data are in Table S4

Table 3: underlying data are in Table S12

 Please review your reference list to ensure that it is complete and correct.

->We have reviewed this list and it is complete and correct. There are no changes from the previous version of the manuscript.

---

## [Decision Letter · Decision Letter 1]

29 Sep 2023

The PitA protein contributes to colistin susceptibility in Pseudomonas aeruginosa

PONE-D-23-24646R1

Dear Dr. Lamont,

We’re pleased to inform you that your manuscript has been judged scientifically suitable for publication and will be formally accepted for publication once it meets all outstanding technical requirements.

Kind regards,

Iddya Karunasagar

Academic Editor

PLOS ONE

Additional Editor Comments (optional):

All reviewer comments have been addressed.

Reviewers' comments:

Reviewer's Responses to Questions

**Comments to the Author**

1. If the authors have adequately addressed your comments raised in a previous round of review and you feel that this manuscript is now acceptable for publication, you may indicate that here to bypass the “Comments to the Author” section, enter your conflict of interest statement in the “Confidential to Editor” section, and submit your "Accept" recommendation.

Reviewer #1: All comments have been addressed

2. Is the manuscript technically sound, and do the data support the conclusions?

Reviewer #1: (No Response)

3. Has the statistical analysis been performed appropriately and rigorously? 

Reviewer #1: (No Response)

4. Have the authors made all data underlying the findings in their manuscript fully available?

Reviewer #1: (No Response)

5. Is the manuscript presented in an intelligible fashion and written in standard English?

Reviewer #1: (No Response)

6. Review Comments to the Author

Reviewer #1: (No Response)

7. PLOS authors have the option to publish the peer review history of their article (what does this mean?). If published, this will include your full peer review and any attached files.

Reviewer #1: No

---

## [Editor Report · Acceptance letter]

4 Oct 2023

PONE-D-23-24646R1 

The PitA protein contributes to colistin susceptibility in *Pseudomonas aeruginosa*

Dear Dr. Lamont:

I'm pleased to inform you that your manuscript has been deemed suitable for publication in PLOS ONE. Congratulations! Your manuscript is now with our production department. 

Kind regards, 

on behalf of

Dr. Iddya Karunasagar 

Academic Editor

PLOS ONE